# Nanotechnology in the Diagnostic and Therapy of Hepatocellular Carcinoma

**DOI:** 10.3390/ma15113893

**Published:** 2022-05-30

**Authors:** Florin Graur, Aida Puia, Emil Ioan Mois, Septimiu Moldovan, Alexandra Pusta, Cecilia Cristea, Simona Cavalu, Cosmin Puia, Nadim Al Hajjar

**Affiliations:** 1Department of Surgery, University of Medicine and Pharmacy “Iuliu Hatieganu”, 400012 Cluj-Napoca, Romania; florin.graur@umfcluj.ro (F.G.); drpuia@yahoo.fr (C.P.); na_hajjar@yahoo.com (N.A.H.); 2Regional Institute of Gastroenterology and Hepatology “Octavian Fodor”, 400394 Cluj-Napoca, Romania; septimiu1995@yahoo.com; 3Department of General Practitioner, University of Medicine and Pharmacy “Iuliu Hatieganu”, 400347 Cluj-Napoca, Romania; 4Department of Analytical Chemistry, University of Medicine and Pharmacy “Iuliu Hatieganu”, 400347 Cluj-Napoca, Romania; alexandrapusta@gmail.com (A.P.); ccristea@umfcluj.ro (C.C.); 5Department of Medical Biophysics, Faculty of Medicine and Pharmacy, University of Oradea, 410087 Oradea, Romania; simona.cavalu@gmail.com

**Keywords:** hepatocellular carcinoma, nanomedicine, cancer diagnosis, targeting, smart therapy

## Abstract

Hepatocellular carcinoma is the most common liver malignancy and is among the top five most common cancers. Despite the progress of surgery and chemotherapy, the results are often disappointing, in part due to chemoresistance. This type of tumor has special characteristics that allow the improvement of diagnostic and treatment techniques used in clinical practice, by combining nanotechnology. This article presents a brief review of the literature focused on nano-conditioned diagnostic methods, targeted therapy, and therapeutic implications for the pathology of hepatocellular carcinoma. Within each subdomain, several modern technologies with significant impact were highlighted: serological, imaging, or histopathological diagnosis; intraoperative detection; carrier-type nano-conditioned therapy, thermal ablation, and gene therapy. The prospects offered by nanomedicine will strengthen the hope of more efficient diagnoses and therapies in the future.

## 1. Introduction

Hepatocellular carcinoma (HCC) is ranked fifth among the most common neoplasms and ranked third among the causes of cancer-related deaths. The incidence of HCC is on the rise in Europe and the USA, due to new lifestyle-related risk factors, such as metabolic syndrome and nonalcoholic fatty liver disease [1,2]. The estimated five-year survival rate of HCC is only 18%, making it one of the most lethal cancers [3]. Thus, it is vital that this disease is diagnosed early and treated adequately, to ensure the best outcomes for the patient. Current diagnostic and treatment methods present certain limitations. The most commonly employed methods for the diagnosis of HCC are imaging techniques (ultrasound, computed tomography, or magnetic resonance imaging) and the quantification of biomarkers that are overexpressed in this pathology, such as alpha-fetoprotein (AFP) [1]. However, imaging equipment might not be available in all clinical facilities, and the results obtained are dependent on the user’s expertise, while AFP poses certain sensitivity and selectivity issues [4]. Treatment methods for HCC include resection surgery, liver transplantation, chemotherapy, radiofrequency ablation (RFA), or transarterial chemoembolization (TACE) [3]. Resection surgery or liver transplantation are indicated in a very small number of patients, and systemic chemotherapy has moderate results (with high recurrence and multidrug resistance [MDR]) and significant side effects. Local therapies, such as RFA or TACE, have only a palliative role [5,6].

The marked increase in the number of patients diagnosed with liver cancer has led to the development of innovative diagnostic and therapeutic technologies, including nanomedicine. Its development was made possible by a more advanced understanding of the mechanisms of liver cancer, and by the collaboration of teams of doctors, chemists, materials specialists, and physicists, who have focused their research on this field. Although surgery and chemotherapy are still the standard treatments for HCC, the outlook suggests that new nano-conditioned therapies will be useful in clinical practice.

Owing to their intrinsic or artificially designed properties, nanomaterials find applications both in the diagnosis and treatment of hepatocellular carcinoma. Their unique properties and functionalities are due in part to their size (between 1 and 100 nm), and among them are increased biocompatibility, ability to detect early HCC, the possibility of use in imaging and diagnosis, and their use as genic or cytostatic transporters. In the context of diagnosis, nanomaterials can be employed for the functionalization of sensor surfaces, increasing the sensitivity of detection for certain biomarkers, or they can be used as tracers for imaging diagnosis. In the context of therapy, nanomaterials can be functionalized with ligands such as antibodies or aptamers that are specific for tumor cells, thus facilitating targeted transport of drugs directly to the tumor site, reducing non-specific toxicity and chemotherapy resistance. Moreover, nanomaterials can be employed for theragnostic approaches, a promising strategy that allows simultaneous therapy and diagnosis [7].

Given the burden of cancer on public health, many nanotechnology-based strategies have been developed for the treatment of cancer. The evolution of nanotechnology for medical applications in cancer is shown in Figure 1.

The main types of nanomaterials used in the treatment of cancer are carbon-based nanomaterials (nanotubes, fullerene, and graphene), lipid-based nanoparticles (liposomes), metallic nanoparticles (gold nanoparticles), superparamagnetic iron oxide, silica-based nanoparticles, polymers, quantum dots, dendrimers, nanoshells, etc.

In this review, the most recent advances in the use of nanomaterials in the diagnosis and treatment of HCC will be presented. Firstly, the characteristics of the main types of nanomaterials will be, briefly, discussed, followed by the presentation of relevant examples of nano-mediated diagnostic (imaging, serologic, histopathologic, and intra-operatory) and therapeutic approaches. Finally, relevant examples of theragnostic approaches for HCC will be provided, and the future perspectives of nanomaterial applications in HCC management will be presented.

## 2. Types of Nanoparticles Used for Diagnosis and Treatment

In recent years, numerous types of nanoparticles have been employed for the diagnosis and therapy of cancers in general and HCC in particular. These nanoparticles can be classified into carbon-based nanomaterials (nanotubes, fullerene, and graphene), lipid-based nanoparticles (liposomes), metallic nanoparticles (gold nanoparticles), superparamagnetic iron oxide, silica-based nanoparticles, polymers, micelles, dendrimers, and virus-like particles (Figure 2). Each class presents unique properties as well as characteristic advantages and disadvantages and can be obtained using different synthesis methods (Table 1). Since the aim of this study was to highlight the medical applications of nanoparticles for HCC diagnosis and treatment, the following section only, briefly, details characteristics of the most common types of nanoparticles used for the targeted delivery or diagnosis of cancer. For a more comprehensive view on their synthesis and characteristics, the reader is referred to other works of the literature [8,9,10,11,12,13].

### 2.1. Carbon-Based Nanomaterials

Numerous forms of carbon, such as carbon nanotubes (CNT), graphene (GRP), fullerenes (FUL), or quantum dots have, recently, gained increasing attention from the scientific community. These nanomaterials have been employed for the targeted delivery of drugs or for cancer diagnosis. However, there is still concern about their toxicity for both humans and the environment [14], so most of their cancer-related applications are limited to their use in the development of sensors for cancer biomarkers. Their high surface area and excellent electrical conductivity [15] make them suitable candidates for the development of the electrochemical sensors used for the detection of biomarkers that are expressed in HCC [16,17,18].

### 2.2. Liposomes

Liposomes are phospholipid vesicles that are comprised of two distinct regions: a hydrophillic core and a lipophillic lipid bilayer, thus being able to incorporate both hydrophillic and hydrophobic drugs [11,19,20]. Liposomes represent a promising carrier for targeted delivery, with many liposome-based drug formulations already on the market. Some of the shortcomings of liposomes, such as their increased uptake by immune cells, can be overcome by modifying their surface with polymers, such as polyethyleneglycol (PEG), or with targeting agents, such as antibodies or aptamers [19]. Liposomes are among the most commonly used carriers in the targeted treatment of cancers, with many examples of liposomes being used in the treatment of HCC. Doxorubicin was most commonly encapsulated into different types of liposomes [21,22,23] as well as sorafenib and short interfering RNA (siRNA) [24].

### 2.3. Metallic Nanoparticles

Metallic nanoparticles, such as gold (AuNP), silver (AgNP), or platinum (PtNP), can be used for both diagnostic and therapeutic purposes.

Gold nanoparticles (AuNPs) are, usually, synthesized by chemical reduction and stabilization using a capping agent, electrochemical deposition on electrode surfaces (in the case of nanoparticles-based electrochemical sensors), or newly emerging biological-synthesis techniques [10,25]. The resulting gold nanoparticles can be easily functionalized, can be used for hyperthermia applications, and can easily be controlled regarding their size and surface properties [11]. Moreover, given their size-dependent visible-light-absorption behavior, their trajectories into cells can be tracked [11]. Although AuNPs are among the least toxic metallic nanoparticles, some studies have shown that high concentrations of AuNPs can, potentially, be genotoxic [26]. Further studies are needed to assess the biocompatibility and long-term effects of AuNPs. Until then, AuNPs and other metallic nanoparticles remain valuable candidates for diagnostic techniques, such as the electrochemical detection of different biomarkers. In the context of HCC, gold nanoparticles [17], gold–platinum [16], or platinum–palladium nanostructures [18] have been used in conjunction with carbon-based nanomaterials, for the detection of biomarkers such as AFP or glypican-3. Gold nanostructures can, also, be employed for the imaging and photothermal ablation of tumors [27].

### 2.4. Magnetic Nanoparticles

Superparamagnetic iron oxide nanoparticles (SPIONs) can be, successfully, employed for both diagnostic [28] and therapeutic applications [29]. These nano-carriers consist of a magnetic core and a coating. Their main advantage consists of their ability to be guided using an external magnetic field, thus being promising candidates for targeted drug delivery. Moreover, magnetic nanoparticles can be used as contrast agents in magnetic resonance imaging diagnosis, due to their superparamagnetic properties [30]. Magnetic nanoparticles can be obtained using a variety of different physical or chemical synthesis strategies and can be functionalized using inorganic or organic ligands to increase their stability [11]. Their versatility and high stability make them desirable nanomaterials for drug delivery and diagnosis [31].

### 2.5. Polymeric Micelles

Polymeric micelles are, also, promising candidates for targeted drug delivery applications. These nano-carriers consist of self-assembled amphiphilic block-copolymers that form a hydrophobic core and a hydrophilic corona [32]. Polymeric micelles can be obtained from a wide variety of polymers, such as poly (lactic-co-glycolic acid), poly (ε-caprolactone), and poly (lactic acid), and are formed when the concentration of the polymer reaches the critical micelle concentration [32]. Polymeric micelles have been employed in the treatment of HCC for the encapsulation of hydrophobic drugs, such as ursolic acid [33] or niclosamide [34], in order to increase their bioavailability.

**Table 1 materials-15-03893-t001:** Synthesis methods, advantages, and disadvantages of different types of nanoparticles.

Nanoparticle Type	Synthesis Methods	Advantages	Disadvantages
Carbon-based NPs	Mechanical exfoliation (GRP) Chemical exfoliation (GRP) Chemical vapor deposition (GRP, CNT) Laser ablation (CNT, FUL) [8]	High surface area High electrical conductivity [15] Chemical stability	Bio-corona formation Toxicity Environmental toxicity [14]
Liposomes	Thin-film method Proliposome method Injection method Emulsification method [9]	Hydrophillic and hydrophobic drug encapsulation Biocompatibility Low imunogenicity	Low encapsulation efficiency [11] Short shelf-life Accelerated blood clearance
AuNPs	Chemical reduction Electrochemical reduction Seed-mediated growth Digestive ripening Biologic synthesis [10]	Easily functionable Hyperthermia applications Visible light absorption High surface area High electrical conductivity [15]	Potential genotoxicity High costs of raw materials
SPIONs	Chemical co-precipitation Thermal decomposition Gas-phase deposition Pulsed laser ablation Electron beam lithography Biologic methods [12]	External guidance Hyperthermia applications High stability	Potential toxicity Laborious synthesis
Polymeric micelles	Phase-inversion method Rehydration method Polymerization-induced self-assembly Microfluid method [13]	Hydrophillic and hydrophobic drug encapsulation Biocompatibility Prolonged circulation time	Low drug loading Low stability [32]

## 3. Nanoparticles Used for Diagnostics

In order to diagnose HCC, nanomedicine can be used in the following directions: imaging diagnosis using computed tomography (CT), magnetic resonance imaging (MRI), real-time up-conversion luminescence imaging, or a combination of these imaging techniques, as well as pre- or intra-operatory diagnosis and serological or histopathological diagnosis, by detecting biomarkers in patients’ biological fluids or biopsy samples.

### 3.1. Imaging Diagnosis

Ultrasound imaging is a valuable tool for the surveillance of patients who are at risk of developing HCC, while CT and MRI are common diagnostic methods for HCC [1]. Although diagnosis is, relatively, specific, due to the existence of the “radiological hallmark of HCC” [1], certain limitations exist in regards to currently employed diagnostic methods. The contrast agents that are, typically, used for imaging diagnosis have short half-lives and do not present specificity for the tumor tissue [35].

Imaging diagnosis can be improved by developing tracers or contrast agents with superior qualities to those currently used by nano-candidacy. Several studies have shown significant progress in the field; however, the properties of these nano-tracers must, additionally, meet certain conditions: have increased stability, so that scanning devices (CT and MRI) can be set in certain parameters, and have comparable image characteristics. Nanoparticles must accumulate in tissues at certain time intervals after injection and should be rapidly eliminated at set time intervals [7]. Due to the versatility of nanomaterials, the majority of imaging approaches for HCC combine their diagnostic properties with therapeutic ones, and, thus, these approaches will be presented in the Section 5, while a few simple imaging diagnostic methods will be presented, herein.

An MRI-detection method for HCC was developed, based on the presence of high concentrations of AFP and glypican-3 in the cytoplasm and on the surface of tumor cells, respectively, compared to healthy cells. Ultra-small SPIONs (5 nm diameter) were functionalized with antibodies for the two biomarkers, by amide bond formation, and the modified SPIONs (SAG) were used for MRI imaging of HCC cells. In parallel, SPIONs functionalized only with AFP (SA) or glypican-3 antibodies (SG) were, also, obtained and tested. Prussian blue staining assay and in vitro MRI tests were carried out to investigate the accumulation of these carriers in tumor cells. The highest Prussian blue staining, as well as the highest reduction in T2 values in MRI, was obtained for SAG compared to SA and SG, indicating that double functionalization can enhance the detection of HCC compared to single functionalization, this being an advantage of the presented method [28].

An aptamer-conjugated fifth-generation dendrimer, for the combined visual near-nfrared fluorescence and MRI imaging of HCC in rats, was proposed. The aptamer was selected for endoglin, a biomarker present on the surface of vascular endothelial cells in tumors, thus increasing the specificity of the nanoprobe towards tumor tissue. The modified dendrimers were labeled with fluorophores, for near-infrared fluorescence detection, and with paramagnetic Gd^3+^-DTPA chelators, which helped achieve T1-weighted relaxation and an MRI signal. The obtained nanoprobes displayed high fluorescence in tumor cells, compared to aptamer-free dendrimers. In vivo MRI imaging showed a clear delimitation of the tumor margins, indicating that this approach could be used for guiding surgical procedures for the resection of small hepatic tumors [35].

Another dual-imaging approach was reported by Lee et al. In their study, Nd^3+^-doped up-conversion nanoparticles (UCNP) were employed for the MRI and real-time up-conversion luminescence imaging of HCC, in a simple rat model. Nd^3+^ was used, due to its lower potential for overheating upon irradiation, compared to the commonly used Yb^3+^. The core-shell particles were obtained via thermal decomposition (the core), followed by shell-coating via a seed-mediated process. The surface of the nanoprobe was functionalized with anti-CD44 antibody, to increase selectivity for the tumor cells. Selective accumulation of the nanoprobes in tumor tissue was demonstrated by MRI scans as well as by the bright UCL signal observed in tumor tissue, compared to the surrounding healthy tissue [36].

Dual MRI and CT for HCC models in rats have, also, been reported. A polyethyleneimine substrate was modified with a gadolinium chelator and lactobionic acid, then used as a scaffold for the deposition of AuNPs. The obtained AuNPs presented selectivity for HCC tissue, due to the lactobionic acid, which acts as a ligand for the asialoglycoprotein receptor present on the surface of HCC cells. The AuNPs presented good biocompatibility, accumulation in the tumor site, and high r_1_ relaxitivity and specificity towards HCC tissue [37].

### 3.2. Serologic and Histopathologic Diagnosis

There is abundant literature on serological biomarkers for the early detection of HCC [38,39,40,41], but relatively few of these biomarkers are used for nano-mediated detection. This may lead to the development of systems that will significantly improve the diagnosis of HCC [7]. The most commonly employed HCC biomarker, and the only one with phase 5 validation data, is AFP, but numerous other biomarkers (glypican-3, osteopontin, Golgi-protein 73, and PIVKA) have been, recently, reported [38].

The levels of serological tumor markers used in the detection and monitoring of HCC are usually low, and the detection of these biomarkers in biological samples requires the development of sensitive methods. Nanomaterials, such as metallic nanoparticles or carbon-based nanomaterials, can be used to increase the surface area of sensors, improve their sensing capabilities, and, thus, increase overall sensitivity. Moreover, they can also be functionalized with biological (antibodies and enzymes) or biomimetic (aptamers and molecularly imprinted polymers) recognition systems, which increase the selectivity of the detection method towards a selected target. Many nanomaterial-based detection systems have been reported for the detection of cancer biomarkers, such as proteins, circulating tumor cells, or miRNA [42,43,44], and some of those related to HCC detection are, briefly, presented herein.

An AFP sensing technique uses an electrochemical method and an AFP aptamer for detection [16]. Aptamers are short, single-stranded nucleic acids that can bind to specific targets [45]. Gold–platinum nanoparticles were deposited on the surface of an electrode, to increase its electrical conductivity, followed by the deposition of reduced graphene oxide, chitosan, and a redox mediator. Lastly, the aptamer was added to ensure selectivity towards AFP. The method showed some interferences with other proteins, but its sensitivity was sufficient for the detection of low concentrations of AFP [16].

Metallic nanoparticles were, also, employed for the electrochemical detection of glypican 3. Li’s group reported two similar methods, using gold nanoparticles [17] and platinum–palladium structures [18], respectively. These nanomaterials were grafted onto electrode surfaces, together with reduced graphene oxide, hemin [17], or ferrocene [18], as redox mediators, and the glypican-3-specific aptamer, which was immobilized via cross-linking. The detection was done by measuring the decrease in the differential-pulse voltammetry oxidation peak of the redox mediator, when glypican-3 was added to the aptasensor. The platinum–palladium nanostructure approach presented higher stability and a better linear correlation coefficient, but its sensitivity was lower, compared to that of the AuNP approach.

A biosensor based on a nanopore structure was used for the multiplexed detection of three cancer biomarkers (AFP, carcinoembryonic antigen, and human epidermal growth factor receptor-2), by charge modulation ion current rectification. The system contained three different antibodies, to ensure selectivity towards each marker. This sensor exhibited excellent sensitivity, with limits of detection in the femtomolar level for undiluted serum samples [46].

### 3.3. Intraoperatory Diagnosis

A new trend that has, recently, emerged is intraoperative diagnosis. This fact refers to the highlighting of the remaining tumor tissue, usually on the hepatic resection trance, which is, often, difficult to visualize without special techniques.

Andreou et al. developed Surface-Enhanced Raman Scattering (SERS) silica-coated nanoparticles, as a contrast agent for the intraoperative detection of HCC in mice. Gold NPs were obtained, using a chemical reduction method, and were, then, coated with silica in the presence of a Raman reporter, using a modified Stöber method. The silica layer was obtained using a non-toxic protocol, ensuring the protection of NPs against aggregation. The detection of HCC, using these NPs, relied on their higher accumulation in healthy liver tissue compared to tumors, thus highlighting the tumor margins. The approach was compared to MRI intraoperatory imaging, and the SERS-based method was able to discern smaller tumors. Moreover, compared to fluorescence methods, the SERS-based approach has higher sensitivity and photostability. One of the limitations of this method is the lack of clinically available Raman imaging devices [47].

Fluorescence-based intraoperatory imaging was, also, reported. Tsunda et al. developed lactosomes, which encapsulated indocyanine green (ICG), a photosensitizing agent for near-infrared fluorescence imaging. Lactosomes are a type of core-shell polymeric micelle, obtained by the polymerization of lactic acid and sarcosine. These NPs were, intravenously, administered to nude mice and demonstrated prolonged retention in liver tumors, compared to ICG alone. This represents an advantage, in the case of long surgical procedures, where a single administration would be sufficient for the whole procedure. Another advantage is the ability of ICG-loaded lactosomes to perform photodynamic therapy of the tumor, upon laser irradiation [48].

Another fluorescence imaging-based nanoconditioned system is that of Dai et al., which is aimed at detecting AFP on the membrane of HCC cells. The system is a complex comprised of CuInS_2_–ZnS quantum dots (QD), gold nanoflowers, and anti-AFP antibody. The QDs were synthesized using a hydrothermal method and were sonicated together with gold nanoflowers to yield Au/CuInS_2_–ZnS, which was later incubated with anti-AFP antibody. The system has the advantage of being biocompatible and has low toxicity. The association between gold and QDs emits higher fluorescence compared to QDs alone, making the complex suitable for in vivo detection. The complex could successfully identify HepG2 HCC cells [49].

## 4. Nanoparticles Used for Therapy

Depending on HCC stage, different treatment strategies can be employed, with various degrees of efficacy and safety for the patient [4]. For very early and early stages of the disease, ablation, resection, or liver transplantation are the methods of choice, in the intermediate stage, chemoembolization is used, and in the advanced stage HCC benefits only from systemic therapy. Nanotechnology can be incorporated into many of these strategies (ablation, chemoembolization, and, especially, systemic drug delivery) in order to overcome their current limitations and provide better alternatives for the patient. Moreover, newly emerging treatment strategies, such as the administration of siRNA or short microRNAs (miRs), which lead to the inhibition of post-transcriptional expression of mRNA targets, have been recently reported [50,51]. Nanotechnology could play a crucial role in the use of these treatments, due to its ability to reduce some limitations of RNA, such as rapid degradation in intra- and extra-cellular environments and low stability.

For nanoconditioned carriers to be used for therapeutic purposes, they must meet certain requirements, which include biocompatibility, rapid degradation and elimination, reduced or, even, absent side effects, reduced immunogenicity, and sufficient stability until the delivery of active drugs.

### 4.1. Nanoparticle Targeting and Drug Release

The accumulation of nano-conditioned systems in the target tissue can be performed passively or actively [52]. Passive accumulation is achieved because of the effect of permeability and increased retention, due to the modified structure of the vascularization of intratumoral neoformation [52]. PEGylated nanoparticles have been, frequently, used for the passive targeting of tumors, due to their high permeability and retention effects [53]. Recent studies have demonstrated correlations between certain properties (hydration state and PEG surface density) of PEGylated dendrimers and their passive-tumor-targeting capacity, thus offering insight into new ways of optimizing synthesis of passive targeting carriers [53]. Passive targeting of tumor cells, using coacervate droplets formed from poly (2-methoxyethyl acrylate) (PMEA), has also been reported in the literature [54]. It was found that the addition of a PMEA solution to water leads to the formation of dense coacervate droplets, which selectively accumulate in certain cancerous cells lines but do not have affinity towards healthy cells. This could provide a facile alternative to the use of ligands for active accumulation.

Active accumulation is induced by the presence on the surface of loaded nanoparticles of ligand molecules, which lead to specific attachment in that tissue. The ligand molecules that can be employed for active targeting are represented by biological (antibodies and enzymes) or pseudo-biological (aptamers and molecularly-imprinted polymers) elements, which can specifically recognize targets that are over-expressed on the surface of cancer cells. Such molecules can be represented by: P-selectin [55], serotonin 1 B and 2 B receptors, gamma-aminobutyric acid A receptors (theta subunit), epidermal growth factor (EGRF), fibroblast growth factor receptor 3 (FGRF3), somatostatin receptor, insulin receptor (present in two isoforms), prostaglandin E2, adeno-self receptors (A2b), insulin-like growth factor, asialoglycoprotein receptor 1 (ASGPR1) for galactosamine [56], folate receptor [56], transferrin receptor [56], AFP, glypcian-3 [56], and retinoic acid receptor-alpha. Specific delivery to the target using this method can reduce the systemic toxicity of active molecules as well as bypass chemotherapeutic resistance systems [57].

Active release mechanisms at the tumor site rely on the use of external or internal factors that trigger drug release, such as ultrasound, pH, temperature, and ionizing radiation. Through these mechanisms, the release of active drugs into the tumor tissue is increased and lowered into healthy tissues.

#### 4.1.1. Systemic Therapy

The structure of the nanoparticles allows their loading with active substances for targeted drug delivery. This feature depends on the ability to make a connection between the nanoparticles and the drug, the porous inner structure, or the outer structure. In some cases, intermediate molecules are attached to the outer surface of the nanoparticles, creating a link between the basic and chemotherapeutic nanoparticles. The use of nanoparticles for the delivery of systemic therapy in HCC can enhance the therapeutic effects of drugs, reduce their systemic toxicity, and overcome some of their physio-chemical limitations. In Table 2, several nanoparticle-based approaches for targeted drug delivery in the treatment of HCC are presented.

The gold standard systemic oncological treatment for hepatocellular carcinoma is sorafenib (SOR), an antiangiogenic agent that binds to receptors on the surface of tumor cells: PDGFR-β, VEGFR 2, and VEGFR 3 [58]. Sorafenib has certain limitations, such as very poor water solubility and low bio-availability [59], which could be overcome by its encapsulation in different nanoparticle-based delivery systems. Different strategies for the encapsulation of SOR in nano-carriers have been reported.

A passive method for SOR delivery was developed using monoolein-based liquid crystalline nanoparticles (LCN). SOR-loaded LCNs were obtained by self-assembly and were, then, coated with polymeric layers of poly-l-lysine (PLL) and polyethylene glycol-b-polyaspartic acid (PEG-b-PAsp), using a layer-by-layer process. The polymeric layers increased the biocompatibility of the particles and contributed to the sustained release of SOR. The release of SOR was higher at pH 5.5, compared to physiological pH, indicating that the method could successfully be used to release SOR in the acidic tumor microenvironment. The SOR-loaded LCNs displayed higher toxicity towards HCC cells, compared to SOR alone [60].

Another passive method relies on the use of SOR-gold nanoconjugates for HCC treatment. Due to concerns related to the safety profiles of metallic nanoparticles, extensive tests were carried out to assess the biocompatibility of the resulted nanoconjugates. No organic solvents were used in the synthesis process, to reduce the potential toxicity of the complex. Biochemical and hematological parameters were determined in rats, after intraperitoneal administration of SOR-Au nanoconjugates, and no signs of toxicity were observed. Moreover, a blinded histological examination of liver, kidney, heart, and brain tissue revealed no pathologic changes after SOR-Au nanoconjuagate administration. The conjugates were tested on tumor cell cultures with induced SOR resistance, and the conjugates demonstrated higher inhibition of tumor cell growth, compared to SOR alone [61].

Active targeting of SOR was achieved, using folate-decorated bovine serum albumin (BSA) nanoparticles loaded with SOR (FA-SOR-BSANP). The nanoparticles were obtained by mixing BSA with an ethanolic solution of SOR, and folic acid was covalently bound on the surface of BSA nanoparticles using NHS/EDC coupling. Cytotoxicity studies demonstrated that FA-SOR-BSANP exhibited higher inhibition of tumor cell growth compared to SOR alone or non-folate-decorated SOR-BSANP. In vivo studies in rats showed that FA-SOR-BSANP had better tumor-targeting properties compared to SOR-BSANP, demonstrating the utility of folic acid as an active targeting ligand [62].

SPIONs have, also, been employed for the active delivery of SOR to tumor cells, using magnetic fields. SPIONs were obtained by the co-precipitation method and functionalized with polyvinyl alcohol [29] or zinc/aluminum-layered double hydroxide [63]. In both approaches, the functionalized nanoparticles demonstrated higher toxicity towards tumor cells and lower toxicity towards healthy hepatocytes, compared to free SOR.

Standard chemotherapeutics used as single agents, such as doxorubicin [64,65] or gemcitabine [66], have produced modest results in the treatment of HCC. Moreover, doxorubicin (DOX), a natural anthracycline, presents severe acute and long-term side effects, including multiple organ toxicity, in a dose-dependent manner [67]. In this regard, scientists have explored many paths to reduce the toxicity related to DOX therapy, without altering its efficacy. One of the many proposed solutions is the encapsulation of DOX in different nanoparticles. A further advantage of this strategy is overcoming chemotherapy resistance, thus rendering doxorubicin more efficient in HCC treatment. In a phase II study [68,69], in which doxorubicin was encapsulated in mixantrone-loaded polybutylcyanoacrylate nanoparticles, patient survival was significantly increased, compared to that of the unencapsulated drug. In another phase II study, in which DOX was encapsulated into poly isohexylcyanoacrylate polymer-based NPs (used as chemoembolizing therapy), significant survival was, also, observed compared to treatment with the native drug [70]. Multiple liposomal formulations of DOX were, also, reported in the literature [21,22,23,55]. ASGPR active targeting was, mostly, employed in these studies and the liposomes were functionalized with a variety of agents to increase circulation time and biocompatibility. Cytotoxicity of the liposomal formulation towards tumor cells was increased, while cytotoxicity towards healthy cells was reduced, compared to free DOX, in all approaches.

The combination of chemotherapeutics with molecules sensitive to various energy fields (ultrasound, magnetic, microwave, etc.) can lead to the development of molecular platforms that release targeted chemotherapy and, in addition, achieve additional therapeutic effects by in situ thermal ablation. The relevance of such therapies is important, especially in the case of tumors with resistance to conventional therapy, in which systemic toxicity is increased.

#### 4.1.2. Nanoparticle-Mediated Nucleic Acid Delivery

Molecular-targeted therapy, based on genes uploaded to NPs, is a relatively new approach. siRNA is a small strain of RNA that can inhibit the expression of pro-oncogenic HCC genes, thus representing a promising treatment alternative. However, siRNAs have a short half-life, due to nuclease degradation in the blood stream [75], and present low bioavailability. Moreover, virus carriers, which are usually used for the transport of siRNAs, can cause immunogenicity and mutagenesis [76].

Biocompatible NPs aim to protect siRNA from degradation, increase bioavailability, and reduce immunogenic effects. Furthermore, by decorating them with molecules that bind, preferentially, to certain targets on the surface of tumor cells, the specificity of antitumor activity may be increased.

Selenium nanoparticles were obtained, using a reduction method, and were employed for the delivery of siRNA in the treatment of HCC. Hyaluronic acid (HA) was used to modify the surface of selenium NPs to increase biocompatibility and, selectively, target HCC cells. Then, positively charged polyethylenimine (PEI) was added onto the particles and anionic siRNA was loaded via electrostatic interaction. The HA-modified SeNPs demonstrated low toxicity, superior tumor accumulation, and apoptosis induction, compared to non-HA modified NPs [76]. A similar approach was developed by the same group, using folic acid (FA) as a targeting ligand for HCC. Similarly, the FA-modified SeNPs exhibited higher cytotoxicity in HepG2 cells and higher anti-tumor efficacy in vivo, compared to non-modified NPs [77].

Supermaramagnetic iron oxide NPs were obtained by co-precipitation and were, also, functionalized with PEI and siRNA and galactose was added as an ASGPR-specific ligand (Figure 3A). The modified SPIONS accumulated at the tumor site in ortothopic mice. Moreover, encapsulation into a nanoparticle system extended the half-life of siRNA [78].

Another system that targeted the ASGPR was developed by Huang et al. Their approach consisted of the development of galactose-modified lipid/calcium/phosphate nanoparticles (LCP NPs), incorporating anti-VEGF siRNA (Figure 3B). The calcium phosphate core of the nanoparticles dissolved into the acidic pH of the tumor microenvironment, thus delivering siRNA. The nano-system demonstrated enhanced gene silencing in vitro and in vivo [75].

Combined delivery of SOR and anti-VEGF siRNA, using pH responsive liposomes, was, also, reported. The liposomes were obtained via thin-film hydration and loaded with the two therapeutic agents. Higher cell internalization was demonstrated for tumor cells in media of pH 6.5, compared to physiological pH, indicating the utility of the prepared liposomes for targeted delivery in cancer tissue. In vitro down-regulation of VEGF was higher for the co-loaded liposomes, compared to SOR alone, siRNA alone, or single-loaded liposomes, indicating promising results [24].

### 4.2. Thermo-Ablation Systems

Thermal ablation systems destroy tumor cells by the thermal effect generated by them after irradiation, with various types of waves, such as those generated by ultrasound and magnetic fields. In situ thermal ablation processes using nanostructures are derived from current radiofrequency ablation therapies or microwave ablation.

Glypican 3-targeting IR780 dye-loaded mesoporous silica NPs were proposed by Ma et al. [79] and were used to produce local hyperthermia upon laser irradiation in vitro and in vivo. Glypican-3 targeting was achieved using a complex protocol. First, chimeric antigen receptor T (CAR-T) cells, targeting glypican-3, were produced by transfection. CAR-T cells are cells that can specifically recognize different tumor-surface antigens, in this case glypican-3. Then, the cells were suspended in lysis buffer and CAR-T membranes were obtained through extrusion. The next step consisted of loading commercially available mesoporous silica NPs with IR780 dye and, then, coating the complex with the obtained membranes by sonication followed by extrusion. Membrane coating ensured selectivity towards glypican-3 and ensured the “stealth” of the NPs in vivo. The nano-complex showed great phototermal effect, due to the presence of IR780 [79].

Wang et al. proposed an NIR (near infrared)-powered synthetic high-density lipoprotein (sHDLs) platform, with which they performed photothermal therapy, with tumor destruction. The synthetic HDLs were obtained through self-assembly, using an ApoA-1 mimetic peptide and a mixture of lipids, and were, then, loaded with photothermal agent DiR and mertansine or vadimezan [80].

Chen et al. developed IR820-PEG-melanin NPs (MNPs) that demonstrated superior penetration of tumor tissues with properties useful in both MRI imaging and in situ tumor ablation. An important advantage of melanin is its biocompatibility, being a pigment that can be naturally found in the human body. The nano-carrier was obtained, by self-assembly of pegylated MNPs and the near-infrared dye IR820, and demonstrated selective tumor ablation in orthotopic mice models [81].

### 4.3. Trans-Catheter Arterial Chemoembolization with NPs

The use of NPs in TACE can lead to the improvement of this technique, by avoiding bead aggregation, extratumoral embolization, inefficiency of small particles, reduced degradation of embolizing agents, and dependence of embolization on vessel diameter, in tumor microcirculation [1].

A recombinant silk-elastin-like protein polymer (SELP) was developed by Poursaid et al., conditioned in the form of a hydrogel that can be injected through a microcatheter, which leads to obstruction of blood flow. In the future, these types of hydrogels can be combined with chemotherapeutics, to increase the antitumor effect of TACE [82].

Other authors have used paclitaxel nanocrystals, to be injected as embolizing agents by the TACE technique. The nanocrystalline structures were obtained by one-pot synthesis through paclitaxel self-assembly and crosslinking with glutaraldehyde. The resulting structures were water soluble, unlike paclitaxel, which displays poor water solubility. Another advantage of the nanocrystals was their intrinsic fluorescence, which could potentially allow real-time monitoring during TACE. Compared to paclitaxel alone, the nanocrystalline assemblies displayed prolonged release of paclitaxel. The structures, successfully, reduced the dimensions of tumor spheroids in vitro. The results of this study are promising, however, further in vivo testing needs to be done, in order to evaluate its safety and efficacy [83].

## 5. Nanoparticles Used for Theragnostic

Theragnostic approaches represent a modern strategy, which combines diagnosis and treatment into a single platform. The development of theragnostic strategies requires interdisciplinary collaborations between different specialists, in order to ensure faster implementation of treatment and better outcomes for the patient [45]. Numerous theragnostic strategies for HCC have been developed.

An example of this mechanism, which includes both diagnosis and treatment, is the system developed by Patra et al., which can be used to track targeted delivery using MRI imaging. The system was composed of SPIONs, which act as an MRI tracer and can, as well, be guided to the tumor site using a magnetic field. The SPIONs were modified with a block polymer for doxorubicin encapsulation and with folic acid for HCC targeting. The release of DOX was achieved, using a pH-dependent mechanism [84].

Another SPION-based approach was developed by Cai et al. In this study, sorafenib was encapsulated into polymer-modified SPIONs and active targeting was achieved using an anti-glypican 3 antibody. The release of sorafenib at the tumor site was done using a double pH and redox-dependent mechanism. The system was also used for MRI imaging of tumors in vivo [85].

Ferrous-platinum nanoparticles were grafted onto montmorillonite (MMT) porous clay to develop a tool for HCC MRI imaging. The magnetic modified-MMT was also loaded with mitoxantrone to endow the complex with therapeutic properties (Figure 4A). Another treatment strategy that was incorporated into this approach was the induction of magnetic hyperthermia upon magnetic field application. The theragnostic MMT complexes were tested in vivo in rats and displayed high tumor inhibition, compared to controls, as well as good tumor-imaging capacity [86].

Another example of theragnostic use of nanoparticles is the one proposed by Wang et al. that combined mesoporous silicon (loaded with doxorubicin) and Au-based Janus particles that show therapeutic effects and qualities of CT-scan contrast tracers [87].

A gold-nanorod micelle complex was loaded with Adriamycin and employed as a theragnostic nano-carrier for HCC. The complex was conjugated with anti-Epithelial cell adhesion molecule (EpCAM) antibodies, to offer selectivity towards HCC stem cells. The nano-carriers were used for the selective destruction of HCC stem cells, by using the effects of Adriamycin-induced cytotoxicity and gold-nanorod-mediated photothermal effect. Upon laser irradiation, higher antitumor efficacy was highlighted, compared to the antitumor action of the complex alone, indicating the synergistic effects of Adriamycin and photothermal effect (Figure 4B). Moreover, the system was able to identify tumors by photoacoustic imaging in an ex vivo model [88].

Palladium nanosheets were, also, employed for photoacoustic imaging, combined with radiotherapy and photothermal effects. Palladium nanosheets were functionalized with iodide radioisotopes, via a simple synthesis method. The resulting modified Pd nanosheets displayed synergistic antitumor activity, and high-quality single photon emission computed tomography (SPECT) images were obtained using this approach in three different in vivo HCC models (Figure 4C) [89].

Carbon-based nanomaterials have, also, been proposed for theragnostic purposes. An antibody specific for glypican-3 was conjugated with reduced graphene oxide and, then, combined with nanobubbles to form a system that was used as an ultrasonic contrast agent, as well as for in vitro photothermal action [90].

Future trends of nanomaterials in drug delivery, such as magnetic nanoparticles (MNPs), have, recently, contributed to important developments in several fields, with a focus on medical applications and oncology. MNPs are widely applied in cancer diagnosis, cancer screening, targeted drug delivery, and cancer treatment and diagnosis [91]. Owing to their magnetism, in particular, MNPs (especially superparamagnetic iron oxide nanoparticles (SPIONs)) have been, mostly, used as contrast agents in cancer screening for magnetic resonance imaging (MRI), computed tomography in magneto-acoustic tomography, and near-infrared imaging. Hybrid nanomaterials have received tremendous interest, owing to the combination of the unique properties of organic and inorganic components in a single material. In this class, magnetic polymer nanocomposites are of particular interest because of their excellent magnetic properties, stability, and good biocompatibility of polymeric films. Several therapeutics based on nanoparticulate polymers and iron oxide drug carriers are, currently, under development. These include polymer–drug conjugates, single-crystal NPs, magnetic nanoclusters, micelles, dendritic polymer carriers, sophisticated stimuli-responsive systems, nanorobots, and diagnostic devices. In addition, a growing number of polymer conjugates and other organic or inorganic drug delivery systems are entering clinical development and evaluation [92].

Carbon-based nanomaterials have been exploited, extensively, in the delivery of therapeutics, due to their easy functionalization of their surface, rendering them capable of crossing biological membranes. Toxicity represents a main concern for their in situ use but, lately, the functionalization with an appropriate targeting ligand aids in the reduction in the cytotoxicity of loaded drugs toward healthy cells and augments therapeutic efficiency [93]. Carbon nanotubes as well as graphene have good mechanical strength, aspect ratio, conductivity, and chemical stability, possessing tunable physical properties, such as biocompatibility and high surface area. However, the lack of solubility in aqueous media, non-homogeneity in size (diameter and length), and the presence of metallic impurities are important drawbacks that, still, limit their use in targeted drug delivery. Moreover, the colloidal instability, the limited synthetic control of graphene or other carbon-based nanomaterials, such as fullerenes and carbon nano-anions, as well as the poor chemical stability in the biological environment and susceptibility to the oxidative environment, are other challenges that have to be overcame in order to expand their area of applications.

Metallic nanoparticles have, as main characteristics, a large surface-area-to-volume ratio as compared to the bulk equivalents, large surface energies, existence as a transition between molecular and metallic states providing specific electronic structure, plasmon excitation, quantum confinement, short range ordering, a large number of low-coordination sites such as corners and edges, a large number of “dangling bonds”, and, consequently, specific and chemical properties as well as the ability to store excess electrons. The Au nanoparticles have, for instance, excellent properties for bioconjugation: they are biocompatible and able to fix a great variety of biomolecules [94].

Nanosomes are promising nanoparticles for drug delivery. Phospholipid nanosomes (small, uniform liposomes) have been developed to overcome some of the limitations associated with conventional liposome manufacturing [95]. Nanosomes can be actively targeted to specific cells by ligand labeling, such as folic acid, transcobalamin, iron transport proteins, hormones, and growth factors, to facilitate their active cellular import. Tumor cells can be passively targeted, to use the effect of retention and permeability.

It appears that drug delivery systems based on nanomaterials hold great potential to overcome some of the barriers to efficient targeting of cells and molecules in inflammation and cancer [96]. In addition, another huge advantage is the possibility to overcome problems of drug resistance in target cells and to facilitate the movement of drugs across barriers, such as those in the brain. However, there are some challenges, such as the precise characterization of molecular targets, the cytotoxicity of some nanomaterials, and the possibility of ensuring that these molecules are expressed only in the targeted organs, to prevent effects on healthy tissues.

## 6. Conclusions

HCC represents the fourth-most common cause of cancer-related death worldwide, with various risk factors. The diagnosis of HCC is, mainly, based on imaging and blood tumor markers detection. HCC is, usually, diagnosed at the clinical advanced stage, so current diagnostic and therapeutic methods do not meet the clinical needs, due to late detection and rapid tumor progressing, therefore, new technologies, drugs, and delivery methods are required in clinical practice.

Significant advances have been made in the field of nanomedicine for the diagnosis and therapy of HCC. The most commonly used nanomaterials for targeted drug delivery were liposomes, but also iron-oxide compounds or organic microspheres, the first being approved by the FDA [7].

Improving the imaging diagnosis of nanotechnology can significantly increase the detection of intrahepatic tumors and decrease the size limit at which these tumors can be identified. In addition, a number of nanostructures have useful properties in the diagnosis of CT, MRI, or ultrasound, and features that make them very useful in the treatment of these tumors, thus leading to the concept of theragnostic (diagnosis and therapy).

The use of nano-conditioned ICG-type intravital tracers can improve the intraoperative evaluation of the extent of liver resection for such tumors, helping surgeons to evaluate hepatic resection performance.

The treatments accessible using nanostructures vary, from the transport of cytostatics to the target tissue, the thermal destruction of tumor cells by activation in certain energy fields, and the introduction of genetic material into tumor cells.

The right diameters, the high specific surface area, and specific physicochemical properties made the nanomaterials important building blocks in imaging diagnosis, serological diagnosis, pathological detection, systematic therapy, thermal ablation, chemoembolization, immunotherapy, and many other aspects for HCC [7].

A number of limitations restrict researchers in using these nanostructures in clinical practice, but, progressively, these molecules will be modified, and their qualities improved; therefore, in the future, an important technological advance is foreseen, and nanomedicine will have a say in anticancer therapies.

## Figures and Tables

**Figure 1 materials-15-03893-f001:**
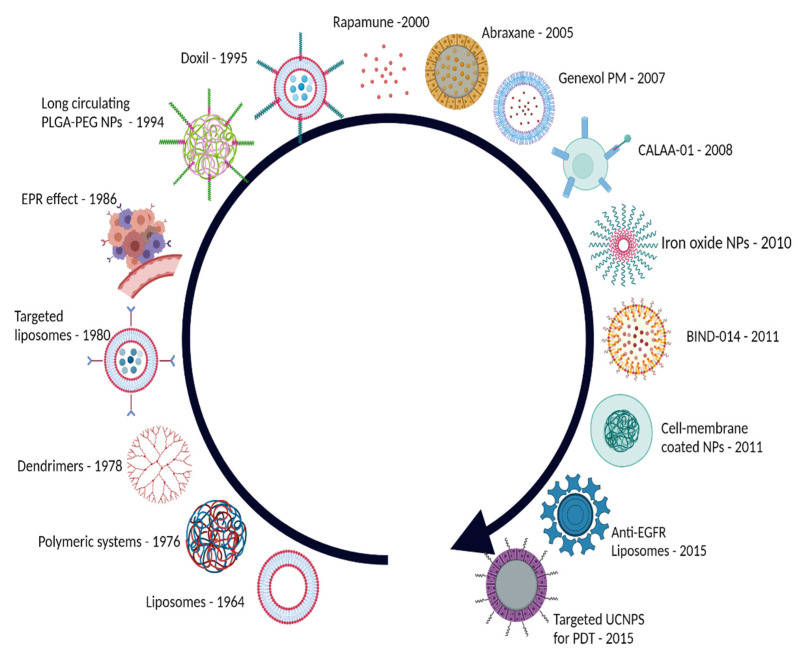
History of nanomedicine. EPR—enhanced permeability and retention; PLGA—poly (lactic-co-glycolic acid); PEG—polyethyleneglycol; UCNPs—upconverting nanoprticles; PDT—photodynamic therapy (created with BioRender.com).

**Figure 2 materials-15-03893-f002:**
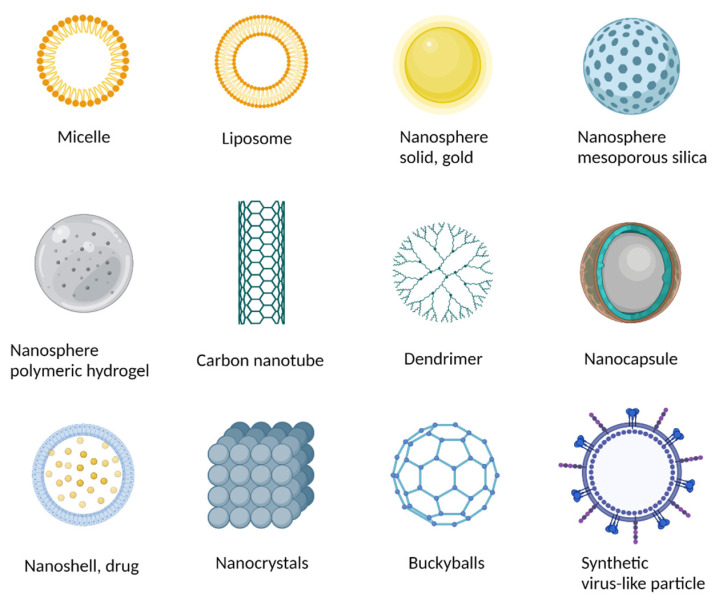
Different types of nanomaterials used in nanomedicine (created with BioRender.com).

**Figure 3 materials-15-03893-f003:**
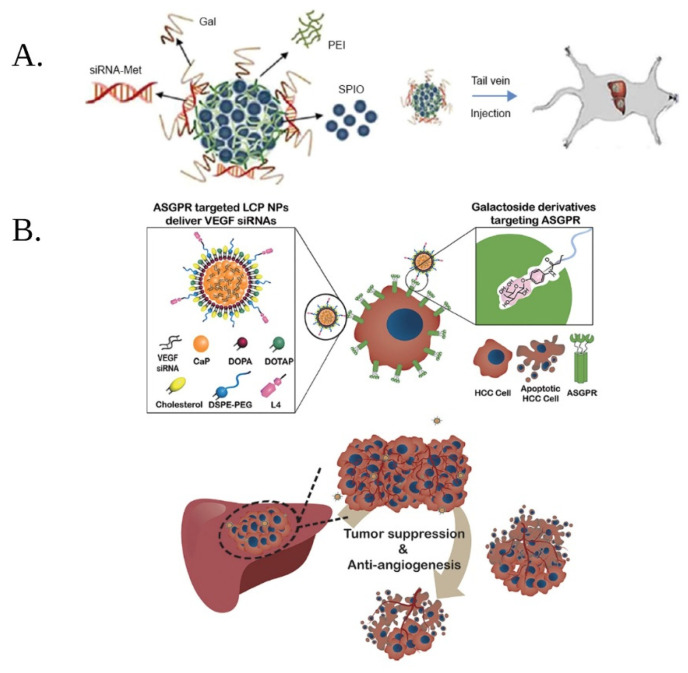
(**A**) Schematic representation of the development strategy for superparamagnetic iron oxide nanoparticles (SPIONs) modified with polyethylenimine (PEI), galactose (Gal), and siRNA as well as their administration in rats, for the treatment of HCC; (adapted from reference [78] (Open Access)); (**B**) Schematic representation of the development strategy of galactose-modified lipid/calcium/phosphate nanoparticles (LCP NPs) incorporating anti VEGF siRNA; DOPA—1,2-dioleoyl-sn-glycero-3-phosphate; DOTAP—1,2-dioleoyl-3-trimethylammonium-propane; DSPE-PEG—1,2-distearoyl-sn-glycero-3-phosphoethanolamine-N-[amino (polyethylene glycol)-2000]; L4—phenyl β-D-galactoside (adapted with permission from reference [75]). (Copyright 2022 American Chemical Society. Created with BioRender.com).

**Figure 4 materials-15-03893-f004:**
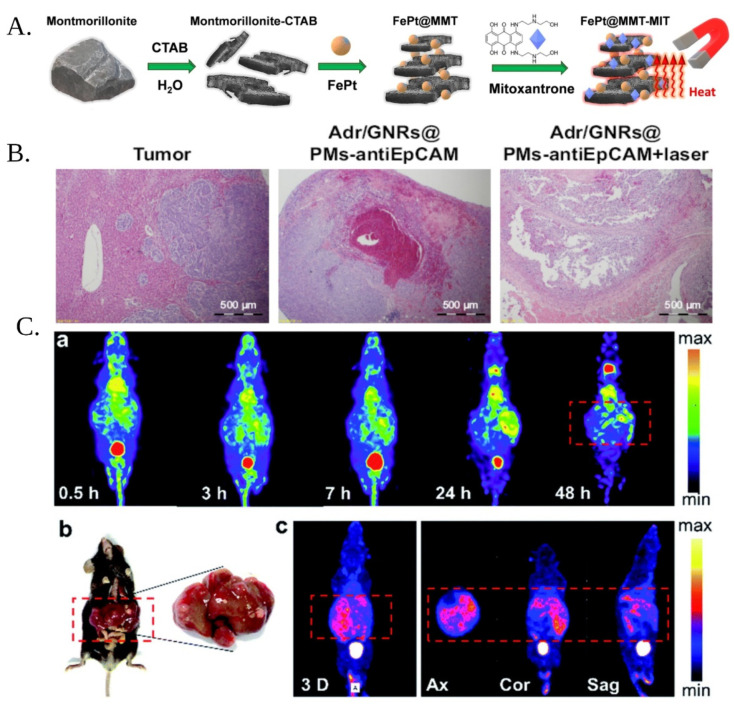
(**A**) Schematic representation of the synthesis of montmorillonite (MMT)-based nanoparticles for theragnostic applications. The MMT reacted with cetyltrimethylammonium bromide (CTAB), to form a layered structure in which iron-platinum nanoparticles and mitoxantrone were incorporated. The platform was used for magnetically induced hyperthermia (adapted from Reference [86] (Open access)). (**B**) Histology studies illustrating the anti-tumor of adryamicin-loaded, gold-nanorod polymeric nanomicelles modified with EpCAM antibody (Adr/GNR@PMs-antiEpCAM), and the same complex under laser irradiation (adapted from Reference [88] (Open Access)). (**C**) SPECT images of the radioactive iodide labeled Pd nanosheets at different time intervals in vivo (**a**) image of liver tissue and (**b**) SPECT images obtained using a conventional tracer used as control method (**c**) (adapted from Reference [89] (Open Access). Created with BioRender.com).

**Table 2 materials-15-03893-t002:** Comparison of different targeted delivery strategies for the treatment of HCC.

Drug	Nanoparticle Type	Diameter (nm)	Targeting	Release	Test Method	Ref.
DOX	Lac-DOPE-L-DOX	96 ± 39	Active (ASGPR ligands)	Cellular uptake of drug	Cell cultures Mice xenografts	[21]
DOX	Lf-PEG-L-DOX	100	Active (ASGPR ligands)	Cellular uptake of drug	Cell cultures Mice xenografts	[22]
DOX	PAG-L-DOX	184.8 ± 1.7	Active (ASGPR ligands)	pH dependent	Cell cultures Mice xenografts	[23]
DOX	Fuc-L-BML-DOX	92.1 ± 12.5	Active (P-selectin ligands)	MW and pH triggered	Cell cultures Mice xenografts	[55]
SOR	LbL-LCN-SOR	165	Passive	pH triggered	Cell cultures	[60]
SOR	FA-SOR-BSANP	158.00	Active (FR ligands)	Cellular uptake of drug	Cell cultures Mice xenografts	[62]
SOR	Lac-SOR/CCM-NPs	115.5 ± 3.6	Active (ASGPR ligands)	pH triggered	Cell cultures Mice xenografts	[71]
SOR	AuNPs-SOR	10	Passive	Cellular uptake of drug	Cell cultures Mice xenografts Ex-vivo studies	[61]
SOR	SPION-PVA-SOR	15	Active (magnetic field)	Magnetic field	Cell cultures Mice xenografts	[29]
SOR	SPION-PEG-ZLDH-SOR	16	Active (magnetic field)	pH dependent	Cell cultures	[63]
GMB	AgNP-GMB	75.1 ± 7	Passive		Cell cultures In vivo toxicity study	[72]
DTX	TPSSNP-DTX	103.6 ± 9.2	Passive	pH and redox triggered	Cell cultures Mice xenografts	[73]
TRP	Gal-Chi-TP-NP	227.4 ± 3.7	Active (ASGPR ligands)	Cellular uptake of drug	Cell cultures Mice xenografts In vivo toxicity studies	[74]

DOX—doxorubicin; Lac—lactobionic acid; DOPE—dioleoylphosphatidylethanolamine; L—liposome; ASGPR—asialoglycoprotein receptors; Lf—lactoferin; PEG—polyethyleneglycol; L—liposome; PAG—palmitoylated arabinogalactan; Fuc—fucoidan; BML—1-butyl-3-methylimidazolium-L-lactate; MW—microwave; SOR—sorafenib; LbL-LCN—layer by layer polymer assembled liquid crystalline nanoparticles; FA—folic acid; BSANP—bovine serum albumin nanoparticles; FR—folate receptor; CCM—curcumin; PVA—polyvinyl alcohol; SPION—superparamagnetic iron oxide nanoparticles; ZLDH—zinc/aluminum layered double hydroxide; GMB—gemcitabine; AgNP—silver nanoparticles; DTX—docetaxel; TPSSNP—D-α-tocopheryl polyethylene glycol 1000-poly (β-amino ester) block copolymer containing disulfide linkages; TRP—triptolide; Gal—galactose; Chi—chitosan.

## Data Availability

Data available upon request from the corresponding author.

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
