# Peer review of "Nanotechnology in the Diagnostic and Therapy of Hepatocellular Carcinoma"

_materials, 2022, doi:10.3390/ma15113893_

Round 1

Reviewer 1 Report

I recommend publication of "Nanotechnology in the diagnostic and therapy of hepatocellular carcinoma" in Materials journal. 

This review article is dealing with the use of nano-medicine for the treatment of Hepatocellular carcinoma. This is the original work of the authors. The manuscript is well written except few typos. The references are up to date. I recommend publication of this work in the Materials journal after corrections of typos.

Author Response

Title: Nanotechnology in the diagnostic and therapy of hepatocellular carcinoma

Manuscript ID: materials-1706570

Answer for reviewer 1:

We would like to thank the reviewer for carefully reading our manuscript and for the kind appreciation of our work. The manuscript was carefully checked and the typos were corrected and highlighted in the revised version of the manuscript.

I recommend publication of "Nanotechnology in the diagnostic and therapy of hepatocellular carcinoma" in Materials journal. 

This review article is dealing with the use of nano-medicine for the treatment of Hepatocellular carcinoma. This is the original work of the authors. The manuscript is well written except few typos. The references are up to date. I recommend publication of this work in the Materials journal after corrections of typos.

Reviewer 2 Report

This manuscript provide a comprehensive review of the strategies of diagnosis and treatment of hepatocellular carcinoma, especially using nanomaterials. The targeting and imaging mechanisms were also well summarized. The strategies and approach for diagnosis and treatment were based on the classification of nanomaterials summarized in Table 1.

  • Polymeric micelles are an essential approach for the development of them. Authors need to mention polymeric micelles
  • The approach of tumor detection and targeting that do not rely on the ligand to a specific receptor expressed on tumor cells have been reported. Authors should summarize the recent advances in tumor detection and targeting, such as the following paper.

 Selective Accumulation To Tumor Cells With Coacervate Droplets Formed From Water-Insoluble Acrylate Polymer, Biomacromolecules, 23(4), 1569–1580 (2022). 

 Different Hydration States and Passive Tumor Targeting Ability of Polyethylene Glycol-modified Dendrimers with High and Low PEG density, Materials Science & Engineering C, 126, 112159 (2021).

Author Response

Title: Nanotechnology in the diagnostic and therapy of hepatocellular carcinoma

Manuscript ID: materials-1706570

Answer for reviewer 2:

We would like to thank the reviewer for carefully reading our manuscript and for providing useful suggestions that will improve the quality of our work. We have addressed each suggestion presented by the reviewer.

This manuscript provide a comprehensive review of the strategies of diagnosis and treatment of hepatocellular carcinoma, especially using nanomaterials. The targeting and imaging mechanisms were also well summarized. The strategies and approach for diagnosis and treatment were based on the classification of nanomaterials summarized in Table 1.

  • Polymeric micelles are an essential approach for the development of them. Authors need to mention polymeric micelles

Information about polymeric micelles was added into Section 2 of the manuscript, as well as in Table 1. The newly-added information was highlighted in the revised version of the manuscript.

  • The approach of tumor detection and targeting that do not rely on the ligand to a specific receptor expressed on tumor cells have been reported. Authors should summarize the recent advances in tumor detection and targeting, such as the following paper.

 Selective Accumulation To Tumor Cells With Coacervate Droplets Formed From Water-Insoluble Acrylate Polymer, Biomacromolecules, 23(4), 1569–1580 (2022). 

 Different Hydration States and Passive Tumor Targeting Ability of Polyethylene Glycol-modified Dendrimers with High and Low PEG density, Materials Science & Engineering C, 126, 112159 (2021).

We would like to thank the reviewer for this suggestion. The content of the two papers was briefly summarized in Section 4.1, which discusses passive targeting of tumor cells. The information was highlighted in the text.

Reviewer 3 Report

The work presents the use of nanotechnology in the diagnosis and therapy against HCC, a carcinoma of importance. The topic is extremely relevant, but discussed in an extremely superficial way. In the sections, details are not given on how the nanoparticles work in the various examples presented, and often the type of nanoparticle itself is not mentioned. Thus, for this review to be acceptable, a reorganization of the data must occur, bringing the action of nanoparticles in detail in all topics. Therefore, it is not possible to accept this article for publication.

For me, part 2 does not bring anything in relation to the central theme of the review, since the references are not related to cancer, and even less to the HCC type. This section must be completely revised.

In section 3, it should be explained how NPs help in CT, PET, MRI, ultrasound, etc...

In the paragraph starting at line 150, which nanotrace and which nanoparticle is used? It seems to me more the description of something, than the discussion of the use of NP itself...

"Ultra-small SPIONs were functionalized with antibodies for the two biomarkers and were used for MRI imaging of HCC cells." How does this process take place, is it advantageous in relation to what is used?

In section 3.1. Lack of depth, authors should discuss the action of NPS in detail and not just cite them. The section must be rewritten.

How does AFP work? This NP assists what in the biomarking of HCC?

"Nanomaterials are used to increase the surface area of ​​sensors, improve their sensing capabilities and thus increase overall sensitivity. Moreover, they can also be functionalized with biological or biomimetic recognition systems which increase the selectivity of the detection method towards a selected target. Several nanomaterials -based detection systems have been reported in the literature." Which nanomaterials? With which biomarkers? What are the works of literature?

"An immunosensor used for fluoroscopic detection of AFP was proposed by Dadfar et al. [23], which used an anti-AFP antibody attached to the surface by the biotin-streptavidin-biotin sandwich technique." Which nanoparticle is used? What is the relevance?

Section 3 needs to be redone, it's very poorly written.

"Moreover, newly-emerging treatment strategies, such as the administration of short interfering RNA (siRNA) or short microRNAs (miRs), which lead to the inhibition of post-transcriptional expression of mRNA targets have been recently reported." Reported where? What's the reference?

The paragraphs in section 4.1 must be reordered, there are no paragraphs with 3 lines...

"Active accumulation is induced by the presence on the surface of nanomedicines of ligand molecules that lead to specific attachment in that tissue." Which nanomedicine? Which particle?

In section 4.1, if the article is about nanomedicine, the nanoparticles should be the target of the discussions, not the ligands... How were the particles synthesized? what particle is used? How did the nanoparticle help? How was the functionalization of the particle performed? What's the end result? These questions must be answered...

Section 4 should be rewritten, as it does not present ANYTHING about the use of nanoparticles... In addition, several paragraphs lack references.

The only section that presents quality in the presentation of results on the use of nanoparticles is section 5.

With the structure of the text presented, it is impossible to infer the results presented in section 6.

The conclusions section is extremely poor.

Author Response

Title: Nanotechnology in the diagnostic and therapy of hepatocellular carcinoma

Manuscript ID: materials-1706570

Answer for reviewer 3:

We thank the reviewer for carefully reading our manuscript and for providing useful suggestions. Each suggestion was addressed and we hope that the changes made will help improve the quality of our work.

The work presents the use of nanotechnology in the diagnosis and therapy against HCC, a carcinoma of importance. The topic is extremely relevant, but discussed in an extremely superficial way. In the sections, details are not given on how the nanoparticles work in the various examples presented, and often the type of nanoparticle itself is not mentioned. Thus, for this review to be acceptable, a reorganization of the data must occur, bringing the action of nanoparticles in detail in all topics. Therefore, it is not possible to accept this article for publication.

For me, part 2 does not bring anything in relation to the central theme of the review, since the references are not related to cancer, and even less to the HCC type. This section must be completely revised.

Section 2 was intended as a general part, providing an overwiev of the properties, advantages and disadvantages of the most commonly used nanoparticles for the diagnosis and treatment of cancers, in general. The section was revised and references regarding HCC were added. A new section bierfly detailing polymeric micelles was added as per another reviwer’s suggestion.

In section 3, it should be explained how NPs help in CT, PET, MRI, ultrasound, etc...

We would like to thank the reviewer for this observation. More information was added in section 3.1 about the way in which NPs were used in MRI and other imaging techniques. The aim of the review was not to provide very detailed explanations, but to create an overview of the possibilities that NPs offer for diagnosis and treatment of HCC. The newly-added information was highlighted in the revised version of the manuscript.

In the paragraph starting at line 150, which nanotrace and which nanoparticle is used? It seems to me more the description of something, than the discussion of the use of NP itself.

In line 150, there is a general discussion about nanoparticles in imaging diagnosis. It does not refer to a specific nanoparticle, but to the fact that the limitations of curently used contrast agents can be partially overcome by developing new nanoparticle-based tracers.

"Ultra-small SPIONs were functionalized with antibodies for the two biomarkers and were used for MRI imaging of HCC cells." How does this process take place, is it advantageous in relation to what is used?

More information was added in this paragraph to explain how antibody immobilization was achieved and what tests were carried out to evaluate the internalization of the SPIONs into tumor cells and thus, to evaluate the capacity for HCC diagnosis, based on this internalization. The main advantage of this technique is that it uses two different antibodies, for two different targets, thus increasing the capacity to detect HCC cells, compared to a traditional, single-antibody approach. The paragraph war rewritten as follws:

„An MRI detection method for HCC was developed based on the presence of high concentrations of AFP and glypican-3 in the cytoplasm and on the surface of tumor cells respectively, compared to healthy cells. Ultra-small SPIONs (5 nm diameter) were functionalized with antibodies for the two biomarkers by amide bond formation and the modified SPIONs (SAG) were used for MRI imaging of HCC cells [21]. In parallel, SPIONs functionalized only with AFP (SA) or glypican-3 antibodies (SG) were also obtained and tested. Prussian blue staining assay and in vitro MRI tests were carried out to investigate the accumulation of these carriers in tumor cells. The highest Prussian blue staining, as well as the highest reduction in T2 values in MRI was obtained for SAG compared to SA and SG, indicating that double functionalization can enhance the detection of HCC compared to single-functionalization, this being an advantage of the presented method.”

In section 3.1. Lack of depth, authors should discuss the action of NPS in detail and not just cite them. The section must be rewritten.

We would like to thank the reviewer for this suggestion. The manuscript was revised and more detailed information was added to section 3.1. The newly added information was highlighted in the text and is also included here:

  „Ultrasound imaging is a valuable tool for the surveillance of patients who are at risk of developing HCC, while CT and MRI are common diagnostic methods for HCC [1]. Although diagnosis is relatively specific, due to the existence of the “radiological hallmark of HCC” [1], certain limitations exist in regards to currently employed di-agnostic methods. The contrast agents that are typically used for imaging diagnosis have short half-lives and do not present specificity for the tumor tissue [29].   

Imaging diagnosis can be improved by developing tracers or contrast agents with superior qualities to those currently used by nano-candidacy. Several studies have shown significant progress in the field; however, the properties of these nano-tracers must additionally meet certain conditions: have increased stability so that scanning devices (CT and MRI) can be set in certain parameters and images have comparable characteristics. Nanoparticles must accumulate in tissues at certain time intervals after injection and should be rapidly eliminated at set time intervals [5]. Due to the versatil-ity of nanomaterials, the majority of imaging approaches for HCC combine their di-agnostic properties with therapeutic ones, and thus these approaches will be present-ed in the section Nanoparticles used for theragnostics, while a few simple imaging diag-nostic methods will be presented herein.

 An MRI detection method for HCC was developed based on the presence of high concentrations of AFP and glypican-3 in the cytoplasm and on the surface of tu-mor cells respectively, compared to healthy cells. Ultra-small SPIONs (5 nm diameter) were functionalized with antibodies for the two biomarkers by amide bond formation and the modified SPIONs (SAG) were used for MRI imaging of HCC cells [22]. In par-allel, SPIONs functionalized only with AFP (SA) or glypican-3 antibodies (SG) were also obtained and tested. Prussian blue staining assay and in vitro MRI tests were car-ried out to investigate the accumulation of these carriers in tumor cells. The highest Prussian blue staining, as well as the highest reduction in T2 values in MRI was ob-tained for SAG compared to SA and SG, indicating that double functionalization can enhance the detection of HCC compared to single-functionalization, this being an ad-vantage of the presented method. 

An aptamer-conjugated fifth-generation dendrimer for the combined visual near—infrared fluorescence and MRI imaging of HCC in rats was proposed. The ap-tamer was selected for endoglin, a biomarker present on the surface of vascular endo-thelial cells in tumors, thus increasing the specificity of the nanoprobe towards tumor tissue. The modified dendrimers were labelled with fluorophores for near-infrared fluorescence detection and with paramagnetic Gd3+-DTPA chelators which helped achieve T1-weighted relaxation and an MRI signal. The obtained nanoprobes dis-played high fluorescence in tumor cells compared to aptamer-free dendrimers. In vivo MRI imaging showed a clear delimitation of the tumor margins, indicating that this approach could be used for guiding surgical procedures for the resection of small he-patic tumors [29].

Another dual-imaging approach was reported by Lee et al. In their study, Nd3+-doped upconversion nanoparticles (UCNP) were employed for the MRI and re-al-time upconversion luminescence imaging of HCC in a simple rat model. Nd3+ was used due to its lower potential for overheating upon irradiation compared to the commonly used Yb3+. The core-shell particles were obtained via thermal decomposi-tion (the core) followed by shell-coating via a seed-mediated process.  The surface of the nanoprobe was functionalized with anti-CD44 antibody to increase selectivity for the tumor cells. Selective accumulation of the nanoprobes in tumor tissue was demon-strated by MRI scans as well as by the bright UCL signal observed in tumor tissue compared to the surrounding healthy tissue [30].    

Dual MRI and CT for HCC models in rats have also been reported. A polyeth-yleneimine substrate was modified with a gadolinium chelator and lactobionic acid and used as a scaffold for the deposition of AuNPs. The obtained AuNPs presented selectivity for HCC tissue due to the lactobionic acid which acts as a ligand for the asi-aloglycoprotein receptor present on the surface of HCC cells [31]. The AuNPs pre-sented good biocompatibility, accumulation in the tumor site, high r1 relaxitivity and specificity towards HCC tissue.” 

How does AFP work? This NP assists what in the biomarking of HCC?

Alpha-fetoprotein (AFP) is a protein that is overexpressed in HCC and can be used as a biomarker for this disease. It is a biomarker approved by FDA, together with its variant AFP-L3, (https://clinicalproteomicsjournal.biomedcentral.com/articles/10.1186/1559-0275-10-13/tables/1) for the screening of HCC. Nanoparticle-mediated detection of AFP could help the early detection of HCC due to NPs ability to increase sensitivity of the detection method, for example in the case of electrochemical detection methods that are described in section 3.2.

"Nanomaterials are used to increase the surface area of sensors, improve their sensing capabilities and thus increase overall sensitivity. Moreover, they can also be functionalized with biological or biomimetic recognition systems which increase the selectivity of the detection method towards a selected target. Several nanomaterials -based detection systems have been reported in the literature." Which nanomaterials? With which biomarkers? What are the works of literature?

More information was added to the paragraph to provide clearer information. We would like to thank the reviewer for mentioning the lack of references in this paragraph. References were added and all the changes were highlighted in the revised text.  

„Nanomaterials such as metallic nanoparticles or carbon-based nanomaterials can be usedto increase the surface area of sensors, improve their sensing capabilities and thus increase overall sensitivity. Moreover, they can also be functionalized with biological (antibodies, enzymes) or biomimetic (aptamers, molecularly imprinted polymers) recognition systems which increase the selectivity of the detection method towards a selected target. Many nanomaterial-based detection systems have been reported for the detection of cancer biomarkers such as proteins, circulating tumor cells or miRNA [36–38]and some of those related to HCC detection are briefly presented herein.”

"An immunosensor used for fluoroscopic detection of AFP was proposed by Dadfar et al. [23], which used an anti-AFP antibody attached to the surface by the biotin-streptavidin-biotin sandwich technique."

Reference [23] was removed from the manuscript as it did not fit the scope of the paper. We thank the reviewer for noticing this mistake.

Section 3 needs to be redone, it's very poorly written.

We thank the reviewer for this observation. Section 3 was revised and changes were highlighted in the revised version of the manuscript.

"Moreover, newly-emerging treatment strategies, such as the administration of short interfering RNA (siRNA) or short microRNAs (miRs), which lead to the inhibition of post-transcriptional expression of mRNA targets have been recently reported." Reported where? What's the reference?

We would like to thank the reviewer for pointing out the absence of references. The references were added into the text of the revised manuscript.

The paragraphs in section 4.1 must be reordered, there are no paragraphs with 3 lines...

One paragraph was deleted and two other paragraphs were merged. Following the suggestion of another reviewer (see below), more information about passive targeting was added into section 4.1.

„The approach of tumor detection and targeting that do not rely on the ligand to a specific receptor expressed on tumor cells have been reported. Authors should summarize the recent advances in tumor detection and targeting, such as the following papers: Selective Accumulation To Tumor Cells With Coacervate Droplets Formed From Water-Insoluble Acrylate Polymer, Biomacromolecules, 23(4), 1569–1580 (2022) and Different Hydration States and Passive Tumor Targeting Ability of Polyethylene Glycol-modified Dendrimers with High and Low PEG density, Materials Science & Engineering C, 126, 112159 (2021).”

"Active accumulation is induced by the presence on the surface of nanomedicines of ligand molecules that lead to specific attachment in that tissue." Which nanomedicine? Which particle?

The aim of this paragraph was to explain what active accumulation is, without refering to a specific ligand or type of nanoparticle. The term „nanomedicine” was replaced with „loaded nanoparticle”.  

In section 4.1, if the article is about nanomedicine, the nanoparticles should be the target of the discussions, not the ligands... How were the particles synthesized? what particle is used? How did the nanoparticle help? How was the functionalization of the particle performed? What's the end result? These questions must be answered...

We would like to thank the reviewer for this observation. More details were added into section 4.1 and the changes were highlighted in the updated version of the manuscript. Information regarding synthesis was only briefly presented because the aim of our work was to highlight the biomedical applications of the nanoparticles, given our group’s expertise in sensing (diagnosis) and treatment of HCC, but not in NP synthesis. We chose to present an overview of different NP-based strategies, without presenting detailed information about these.

Section 4 should be rewritten, as it does not present ANYTHING about the use of nanoparticles... In addition, several paragraphs lack references.

We would like to thank the reviewer for this suggestion. More detailed information was added into section 4, the paragraphs were checked and the missing references were added and highlighted. The remaining pragraphs without references contain discussions based on the presented results. 

The only section that presents quality in the presentation of results on the use of nanoparticles is section 5.

We would like to thank the reviewer for the kind appreciation.

With the structure of the text presented, it is impossible to infer the results presented in section 6.

The intention of the authors was to present future trends in the development and applications of nanomaterials in the detection and treatment of cancer, in general, not only in HCC. Section 6 was changed and all the changes were highlighted in the revised manuscript.

The conclusions section is extremely poor.

The section was revised and new information was added. The changes were highlighted in the revised manuscript.

Round 2

Reviewer 3 Report

Although the questions have been answered, the article is still a long way from fitting into a good review of materials applied to HCC. Details are lacking on the synthesis of materials, functionalization procedures, characterizations, correlations between size/shape and activity, among other details. In this way I believe that it cannot be accepted in the Materials journal.

Author Response

Answer for Reviewer 3

We would like to thank the reviewer for the feedback.

Although the questions have been answered, the article is still a long way from fitting into a good review of materials applied to HCC. Details are lacking on the synthesis of materials, functionalization procedures, characterizations, correlations between size/shape and activity, among other details. In this way I believe that it cannot be accepted in the Materials journal.

The text of the manuscript was modified and more detailed information was added into Table 1, sections 3.3, 4.2 and 4.3 to address the reviewer’s suggestions.

Round 3

Reviewer 3 Report

I believe that good changes were made, regarding the materials area, which should be the central theme of the review and the article can be accepted.

This manuscript is a resubmission of an earlier submission. The following is a list of the peer review reports and author responses from that submission.